# Molecular Identification of *Photobacterium damselae* in Wild Marine Fish from the Eastern Mediterranean Sea

Danny Morick [1,2,3,*,†], Yuval Maron [4,†], Nadav Davidovich [1,2,5], Ziv Zemah-Shamir [1,2], Yaarit Nachum-Biala [4], Peleg Itay [1,2], Natascha Wosnick [6], Dan Tchernov [1,2,3] and Shimon Harrus [4]

1   Morris Kahn Marine Research Station, University of Haifa, Haifa 3780400, Israel
2   Department of Marine Biology, Leon H. Charney School of Marine Sciences, University of Haifa, Haifa 3498838, Israel
3   Hong Kong Branch of Southern Marine Science and Engineering, Guangdong Laboratory (Guangzhou), Guangzhou 524057, China
4   Koret School of Veterinary Medicine, The Hebrew University of Jerusalem, Rehovot 76100, Israel
5   Israeli Veterinary Services, Bet Dagan 5025001, Israel
6   Post-Graduation Program in Zoology, Federal University of Paraná, Curitiba 81530-000, Brazil
*   Correspondence: dmorick@univ.haifa.ac.il
†   These authors contributed equally to this work.

**Abstract:** Infectious diseases caused by marine bacterial pathogens inflict increasing economic losses to fisheries and aquaculture, while also posing a growing risk to public health and affected species conservation. In this study, four wild marine fish species were collected at five fishing sites in Israel, divided into two regions—north (Acre, Haifa, Shefayim) and center-south (Tel-Aviv and Ashdod), and screened for *Photobacterium damselae*. An initial screening was carried out using PCR analysis with specifically designed primers on DNA extracted from livers and kidneys. *P. damselae*-positive samples had their 16S rRNA amplicons sequenced. Later, an attempt to specify relevant sub-species was performed, using a three-layered gene screen: *Car*, *ureC* and *toxR*. Of 334 fish samples, 47 (14%) were found to be *P. damselae*-positive, of which 20 were identified as *P. damselae* subsp. *piscicida* (Phdp), two as *P. damselae* subsp. *damselae* (Phdd) and 25 could not be identified to subspecies. Our results strengthen the view that fish residing in a polluted environment are receptive of pathogenic microorganisms. To assess how the presence of pathogens may affect population management and conservation, this research should be followed by studies aimed at: (i) quantifying levels of pollutants that may affect pathogen emergence, and (ii) creating a standard pollution-level index as a basis for setting criteria, above which authorities should take measures of precaution.

**Keywords:** wild fish pathogens; *Photobacterium damselae*; diseases; Mediterranean sea

## 1. Introduction

Fishes are the carriers of many pathogenic agents such as viruses [1], bacteria [2], parasites [3] and fungi [4], some of which may bear a zoonotic hazard [5,6]. Studies performed worldwide directed at identifying pathogenic agents in fish, focus mainly on farmed fish and to a lesser extent on wild populations. This is equally true for the Levantine basin of the Mediterranean Sea. This study is part of the long-term Mediterranean Sea monitoring program, conducted by the University of Haifa, Israel. The program was established in aim of creating a database to aid progressing ecological research, to assess the health status of the marine ecosystem, and to identify important changes which indicate deterioration and damage to the biological system, its diversity and its function. Samples were collected from two geographic regions in Israel: north (Acre, Haifa and Shefayim) and center-south (Tel-Aviv, Ashdod). From each region, four fish species were sampled: round sardinella (*Sardinella aurita*), brushtooth lizardfish (*Saurida undosquamis*), striped red mullet (*Mullus surmuletus*), and goldband goatfish (*Upeneus moluccensis*). These were

chosen due to their ecological importance and the fact that they are widespread and widely consumed by humans [7,8]. This research is part of a wider effort to create a data baseline of pathogenic prevalence in marine fishes along the Israeli Mediterranean coastline, a region in which there is still a large knowledge gap. Other studies performed as part of this program focused on nervous necrosis virus (NNV) and *Streptococcus* spp.—especially on *S. iniae* [9], *Vibrio* spp. and *Mycobacterium* spp. [10], and *S. iniae*, *V. harveyi* and *P. damselae* [11]. All three studies included sampling of kidneys and livers.

This research focused on two potentially pathogenic marine bacteria: *Photobacterium damselae* and *Rickettsia*-like organisms (RLOs). These putative pathogens were recorded in fishes in other studies worldwide, and are known to be significant and widespread causes of disease and mortality in marine animals and humans [12,13]. *P. damselae*, a commensal marine bacterium that is also capable of becoming a pathogenic agent, has two subspecies: *P. damselae* subsp. *damselae* (Phdd) and *P. damselae* subsp. *piscicida* (Phdp). Phdd is a generalist disease-causing agent in a variety of marine animals, including many fish species, crustaceans, molluscs, marine reptiles, elasmobranchs and cetaceans [12,14]. In humans it may cause opportunistic infections, be involved in necrotizing fasciitis and in extreme cases, mortality [15,16]. Strains of this pathogen were isolated from seawater, seaweed, marine organisms bearing no clinical signs and from seafood, and it is considered abundant in the digestive system of certain species of sharks [12,17]. Infection of marine animals can be transmitted through the water and is affected by temperature and salinity levels [18]. Symptoms appear in certain fish species as hemorrhages and ulcers on the skin, around the gills, pelvic fins and caudal peduncle. In other fish species, hemorrhages appear in the eyes, mouth and jaws [19]. Infection of humans was reported mainly following open injuries during work with fish and fish-bearing water [20], but rare reports of infection after consuming raw fish [21] and infections of the urinary system after swimming in polluted water were also reported [22]. Phdp is the cause of pasteurellosis, a bacterial septicemia also known as fish pseudotuberculosis [22]. This is due to the fact that in chronic infections, fish present lumps/whitish granulomas in several internal organs, in which aggregations of bacteria are found. The main surrogates for this disease are marine fishes. It was isolated during many outbreaks in extensive areas around the globe, including the shores of the US, Japan, northern Europe and throughout the Mediterranean Sea [23]. However, reports on wild fishes are relatively few and recent, whereas the majority of the reports over the years are of disease outbreaks in food-fish species [12,14]. Today, pasteurellosis is a primary limiting factor in mariculture, affecting sole, trout, eels, sea bass and more, and has an extensive economic impact throughout the globe due to the losses that it inflicts [23]. The pathways of infection are still unclear, but it seems that the prevalence rises during summer [24], and is affected by the water temperature (above 23 °C) [25], salinity levels [26] and water quality [25]. Studies on the survivability of this bacterium in seawater and sea sediments point both at a short-term survivability in culture far away from fish, and at the ability to go into a vitality sustaining survival mode [27]. The aquatic environment may serve as a reservoir and source of potential infection. Some studies point at a high homology in the *16S rRNA* gene between the two subspecies, although there is a disagreement regarding the percentage of homology. Some reported 100% homology, while others found differences in one or two nucleotides [28]. One study seemed to have been able to discriminate between the two subspecies using PCR enhancement techniques, combining specific primers for the *16S rRNA* and *ureC* genes. The *ureC* gene, which enables Phdd to hydrolyze urea, is absent from Phdp DNA, therefore becoming a marker distinguishing between the two *P. damselae* subspecies. [29]. Another method for discrimination between the two subspecies regards virulence factors. The transmembrane transcriptional activator protein, ToxR, is involved in virulence regulation. The gene responsible for its production, *toxR*, is considered a valuable phylogenetic marker in studies of *P. damselae* species, as it aids in the differentiation between the subspecies Phdd and Phdp due to its high level of divergence [30]. There are phenotypic traits found only in Phdd, including mobility, nitrate reduction and hemolysis of sheep blood agar.

Furthermore, most Phdd strains can grow at 37 °C (an inhibitory temperature for Phdp), a trait potentially enabling the establishment and flourishing of bacterial populations in homeotherms [31].

In addition to the main objective, the detection of *P. damselae*, this study also examined RLOs. The term RLOs refers to a group of Gram-negative, obligatory intracellular bacterial species that resemble *Rickettsia*—not all of which are fully defined and identified. These bacteria express pathogenic capabilities and are becoming ever more prevalent among fish species globally, both fresh- and salt-water residents. These pathogens are associated with disease, have been reported to cause mortality in marine populations and are responsible for extensive damages to aquaculture [13]. The bacterium *Piscirickettsia salmonis* was the first RLO to be identified [32]. The primary damage it inflicted thus far, and which has led to massive economic losses, is on the salmon industry, although reports emerge of variants affecting other fish species and in new areas around the world [33]. The most consistent external clinical symptoms in infected fish are anemia-driven gill pallor, swollen abdomen, petechiae and ecchymoses at the bases of the fins, around the eyes and in the perianal area. Skin lesions in varying levels of severity and severe ulcers often appear in sick fish [34,35]. The infection and damage occur in marine farms as well as freshwater fish ponds. It was found that the survivability of the bacteria outside of the host is affected mainly by temperature and salinity [36]. Possible infection through the skin has been reported, as well as via the gills and digestive system [33,37]. Still, it is not clear whether transmission is possible also via a vector or vertically. In any case, it is assumed that effective transmission happens through direct (skin) contact [38]. Outbreaks occur primarily following the transfer of fish from freshwater ponds to seawater rearing facilities [37]. These facts highlight the economic and ecologic importance of the current study.

## 2. Materials and Methods

### 2.1. Fish Collection

Fish were caught by local fishers using nets or trawlers, during a period spanning August 2016 to December 2017. The fish were caught at five fishing grounds: Acre, the Kishon river estuary, Shefayim, Tel Aviv and Ashdod, and were bought from merchants of the relevant fish-markets. The fishing grounds were divided into the northern grounds (Acre, Kishon, Shefayim) and central-southern ones (Tel Aviv and Ashdod). All fish were externally inspected and seemed healthy. The fish were put on ice immediately after being purchased and taken to the lab at the Morris Khan Marine Research Station (MKMRS), where they were frozen at −80 °C until being dissected.

### 2.2. Sample Collection

Necropsy was performed according to the protocol described by Yanong [39]. The fish were thawed, underwent external inspection of fins, skin and gills, were weighed and measured from nose to tip of the tail, and then internally inspected. All fish appeared to be in general good health, with no apparent internal or external indications of severe illness. During the necropsy, samples of kidneys (n = 166) and livers (n = 168) were taken. We gathered a total 92 samples from brushtooth lizardfish, *Saurida undosquamis* (liver = 47, kidney = 45), 81 from striped red mullet, *Mullus surmuletus* (liver = 41, kidney = 40), 81 from goldband goatfish, *Upeneus moluccensis* (liver = 39, kidney = 42), and 80 from round sardinella, *Sardinella aurita* (liver = 41, kidney = 39). We aimed to have a similar number of samples per species from each region, and in total collected 179 samples from the northern locations and 155 from the central-southern ones (see Table 1). If skin lesions or parasites were observed, these were documented. Tissue samples were frozen at −80 °C until DNA extraction was performed.

**Table 1.** A summary of the samples collected—divided between the regional origins of the fish, the sampled organ and fish species.

| Fish sp. | Organ | North | Central-South | Total |
|---|---|---|---|---|
| striped red mullet | Kidney | 22 | 18 | 40 |
| | Liver | 23 | 18 | 41 |
| round sardinella | Kidney | 26 | 13 | 39 |
| | Liver | 28 | 13 | 41 |
| brushtooth lizardfish | Kidney | 21 | 24 | 45 |
| | Liver | 23 | 24 | 47 |
| goldband goatfish | Kidney | 18 | 24 | 42 |
| | Liver | 18 | 21 | 39 |
| **Total** | | **179** | **155** | **334** |

*2.3. DNA Extraction*

The tissues were processed at Prof. Shimon Harrus's lab at the Koret School of Veterinary Medicine, located at the Rehovot campus of the Hebrew University of Jerusalem. The research focused on the identification of bacterial pathogenic agents using molecular methods. The primers were ordered according to literature-published sequences [28,37,40], as listed below. DNA extraction was carried out using the Wizard® SV Genomic DNA Purification System (Promega, Fitchburg, WI, USA), according to manufacturer instructions.

*2.4. PCR*

For initial screening of *P. damselae*-positive samples we used a set of *16S rRNA* gene primers—P1, P2 (see Table 2). Positive samples underwent a second round of PCR, aimed at discriminating between the two subspecies, Phdd and Phdp. Samples positive for *ureC* were identified as Phdd, based on a phenotypic difference in the ability to hydrolyze urease, one that Phdd is capable of and Phdp is not [29]. Samples negative for *ureC* and positive for *toxR* and/or *Car* were listed as Phdp. The rest, samples found negative for all three primer sets, were listed as unidentified *P. damselae*. PCR reactions were performed in a total volume of 25 µL using the PCR-ready High Specificity mix (Syntezza Bioscience, Jerusalem, Israel); a volume of 0.5 µL for each of the primers (at a concentration of 200 nM); 2 µL DNA and sterile DNase/RNase-free water (Sigma, St. Loius, MO, USA). Amplification was performed using a programmable conventional thermocycler (Biometra, Goettingen, Germany). Initial denaturation performed at 95 °C for two minutes, followed by 40 cycles of 95 °C (30 s), 55 °C (30 s) and 72 °C (30 s). The process was finalized with a single cycle of 72 °C, for five minutes. The primers used for *P. damselae* amplification are specified in Table 2.

For the RLOs *16S rRNA* gene, PCR reactions were performed as described before [37,40]. Briefly, PCR reactions preformed in a total volume of 25 µL, using the PCR-ready High Specificity mix (Syntezza Bioscience, Jerusalem, Israel); a volume of 0.8µL for each of the primers (at a concentration of 320 nM); 2 µL DNA and sterile DNase/RNase-free water (Sigma, St. Loius, MO, USA). Amplification was performed using a programmable conventional thermocycler (Biometra, Goettingen, Germany). Initial denaturation at Initial denaturation performed at 94 °C for 5 min, followed by 35 cycles of 94 °C (30 s), 61 °C (45 s) and 72 °C (30 s). The process was finalized with a single cycle of 72 °C, for five minutes. The PCR products of both RLOs and *P. damselae* were electrophoresed on 1.5% agarose gels stained with ethidium bromide. They were then evaluated under UV light for the size of amplified fragments, by comparison to a 100-bp DNA molecular weight marker (HyperLadder IV, Bioline, London, UK). Positive control and Non-template control (NTC) were included in each run. Positive controls for both *P. damselae* subspecies were obtained from the fish disease laboratory of the veterinary institute of Italy, in Padova. Positive products were sent for sequencing at Hylabs (Rehovot, Israel). Obtained sequences were then edited using Chromas® and MEGA7® (Chromas software version 2.6.5, Technelysium Pty. Ltd., Tewantin, Queensland, Australia; MEGA software version 7.0.26, Penn

State University, PA, USA). The final sequences were determined by BLASTn against the GenBank database.

**Table 2.** Primers used to screen for *P. damselae*, RLOs and identify *P. damselae* subspecies out of *P. damselae*-positive PCR products.

| Primer Name | Sequence | Gene Target | Bacterium | Size of Amplicon | Reference |
|---|---|---|---|---|---|
| P1 | TAGTGTAGTTAACACCTGCAC | *16S rRNA* | | 570 | [28] |
| P2 | ACACTCGAATCTCTTCAAGT | | | | |
| CAR1 | GCTTGAAGAGATTCGAGT | *16S rRNA* | | 267 | [41] |
| CAR2 | CACCTCGCGGTCTTGCTG | | *P. damselae* | | |
| URE-5 | TCCGGAATAGGTAAAGCGGG | *ureC* | | 448 | [30] |
| URE-3 | CTTGAATATCCATCTCATCTGC | | | | |
| ToxR-5 | GGGATTTTATGGTACACAAA | *toxR* | | 985 | [30] |
| ToxR-3 | ATCATAACCAGAGAGATGCT | | | | |
| PS2S | CTAGGAGATGAGCCCGCTTG | *16S rRNA* | RLOs | 469 | [37,40] |
| PS2AS | GCTACACCTGCGAAACCACTT | | | | |

*2.5. Statistical Methods*

2.5.1. Sample Size

The working hypothesis, based on previous studies from the region and the world, was that the prevalence of RLOs within the populations of the studied fish is ~10% [42,43] and that of *P. damselae* is ~20% [44,45]. In order to estimate the rate with a 3% level of accuracy and a 90% confidence rate, the sample size was calculated to be 384 PCR test per pathogen. For reaching a 4% accuracy level and a 90% confidence rate, a 216 PCR test sample size was required (per pathogen). We chose intermediate rates, thus using a 334-test sample size per pathogen.

2.5.2. Data Analyses

In order to test whether there is a difference in pathogen prevalence between the sampled species, the sampled organs, and whether a difference between the sites exists (considering north and central-south), a logistic regression model was applied. The two significant factors (fish species, fishing site) were put into a multifactorial logistic regression model. Both factors remained significant at the same odds ratio (OR) order of magnitude, which confirmed that there is no confounding phenomenon. The data analyses were performed using SPSS version 25, applying a significance level of 95%.

**3. Results**

No RLOs-positive results were retrieved in any of the tested samples. *P. damselae* was detected in 47 samples (a total prevalence of 14%), of which the brushtooth lizardfish had 4/92 (4.3%), striped red mullet had 13/81 (16%), the goldband goatfish—14/81 (17.3%) and the round sardinella—16/80 (20%) positive samples. Further classification of positive samples into subspecies of *P. damselae* revealed that 20 of the 47 positive samples were of Phdp, two were of Phdd, and the other 25 could not be identified to subspecies (Figure 1).

By aligning sequences using MEGA7®, we observed that some of these 47 positive samples exhibited identical sequences, and 20 unique sequences were obtained. Those unique sequences were then compared to the Genbank database using BLASTn, and 18 sequences expressed an identity of >97% to the *16S rRNA* gene of *P. damselae* (accession no. MK064503.1 for Phdd, NR_037019.1 for Phdp). The two other sequences showed 95.9% and 90% identity to Phdd; 95.7% and 89.8% identity to Phdp. We further compared prevalence between samples collected from sites (north/center-south), and between the two organs (kidney/liver). Of the 47 *P. damselae*-positive samples, 20 were found in kidneys (north = 14, south = 6; a pathogen prevalence of 12.6%) and 27 in livers (north = 20, south = 7; a pathogen prevalence of 15.4%). Both Phdd-positive samples were found in round sardinella livers from the north, while 12 of 20 Phdp-positive samples came from

round sardinella originating in the northern region (see Figure 2). Overall, we found that prevalence of pathogens was higher in the northern sampling sites than in central-southern ones (north = 34, south = 13). The 18 sequences with > 97% resemblance to *P. damselae* were submitted to GenBank (see Table 3 for details).

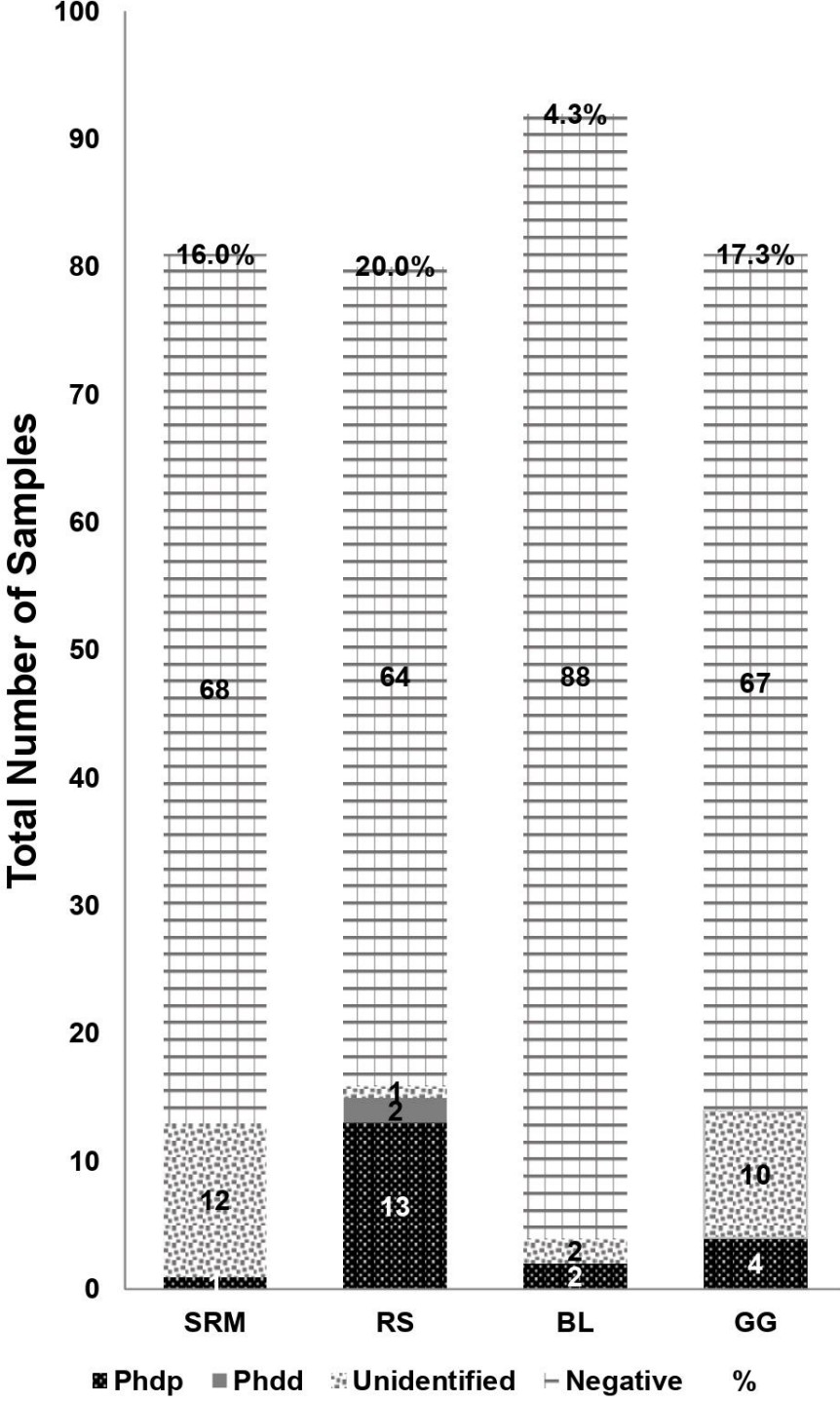

**Figure 1.** Samples divided by fish species: striped red mullet (SRM), round sardinella (RS), brushtooth lizardfish (BL) and goldband goatfish (GG), with depictions of number of samples identified as *P. damselae* subsp. *piscicida* (Phdp), *P. damselae* subsp. *damselae* (Phdd), *P. damselae* with no subsp. (Unidentified) or samples found not to contain *P. damselae* at all (Negative). Percentage refers to the total number of samples identified as *P. damselae* out of the total number of samples.

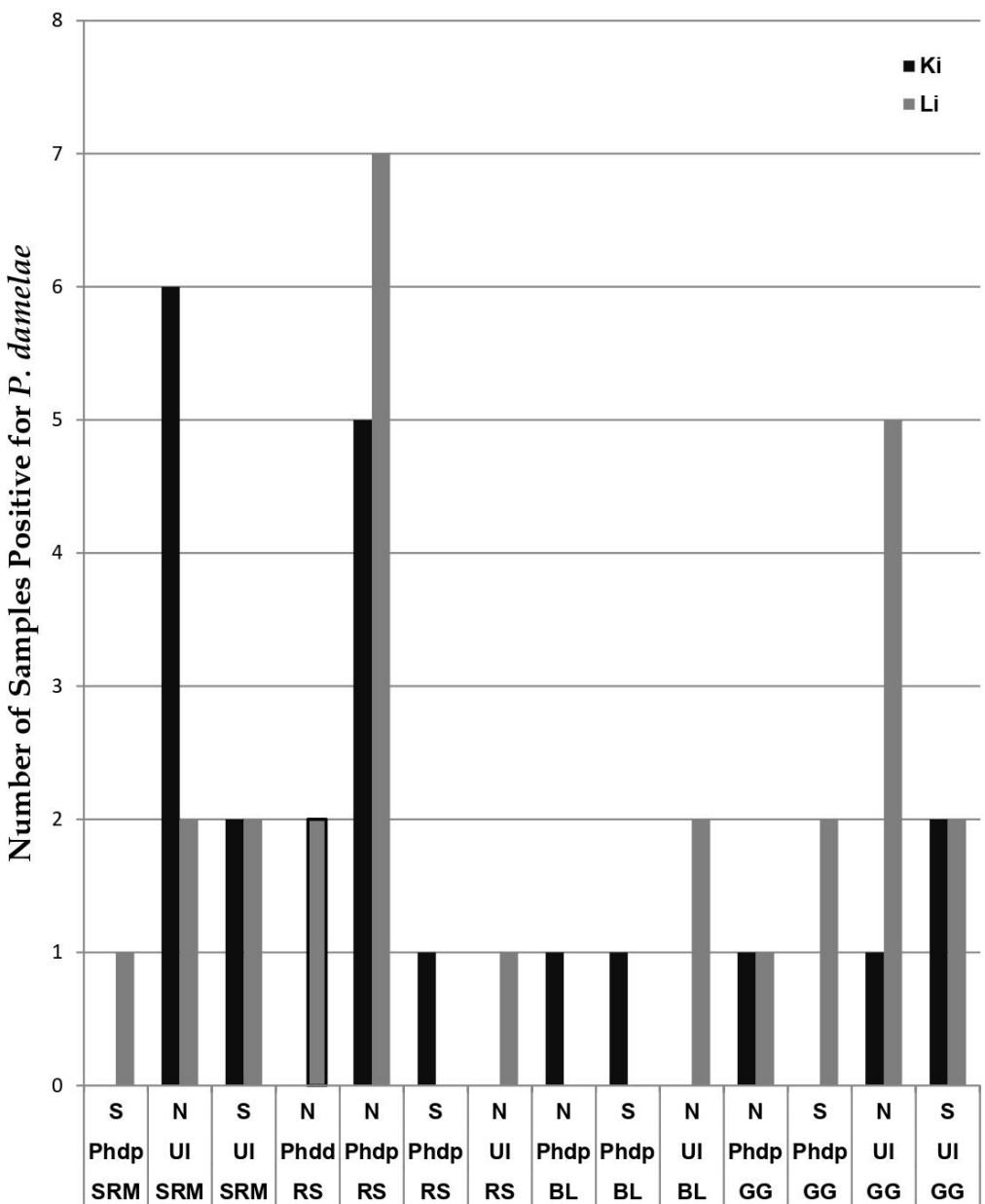

**Figure 2.** Samples divided by fish species: striped red mullet (SRM), round sardinella (RS), brushtooth lizardfish (BL) and goldband goatfish (GG), with depictions of number of samples identified as *P. damselae* subsp. *piscicida* (Phdp), *P. damselae* subsp. *damselae* (Phdd) or *P. damselae* with no subsp. (UI = Unidentified). Samples are further divided by regions of their origins (N = North, CS = Central-South), and by organ they were found in (Ki = Kidney, Li = Liver).

The purpose of the three-gene multiplex PCR screening approach was to try and determine the *P. damselae* subspecies. Five of these sequences were negative for all three genes (*Car*, *ureC* and *toxR*) and therefore could not be unequivocally identified as being of either subspecies. They were thus listed only as *P. damselae*. The two sequences listed as Phdd were positive for *ureC*, while all the rest (i.e., those identified as Phdp) were positive for *Car* and/or *toxR*.

**Table 3.** Accession numbers and details about sequences submitted to GenBank.

| Accession No. | Species | Isolate | Host | Organ | Fishing Source |
|---|---|---|---|---|---|
| OP493090 | *Photobacterium damselae* subsp. *piscicida* | SaK1_191017.11 | round sardinella | Kidney | Kishon estuary |
| OP493091 | *Photobacterium damselae* subsp. *piscicida* | SaK2_191017.15 | round sardinella | Kidney | Kishon estuary |
| OP493092 | *Photobacterium damselae* subsp. *piscicida* | SaK3_191017.18 | round sardinella | Kidney | Kishon estuary |
| OP493093 | *Photobacterium damselae* subsp. *piscicida* | SaK4_191017.5 | round sardinella | Kidney | Kishon estuary |
| OP493094 | *Photobacterium damselae* subsp. *piscicida* | SaK5_211117.21 | round sardinella | Kidney | Tel Aviv |
| OP493095 | *Photobacterium damselae* | SaL1_281117.2 | round sardinella | Liver | Acre |
| OP493103 | *Photobacterium damselae* subsp. *piscicida* | SaL10_281117.4 | round sardinella | Liver | Acre |
| OP493104 | *Photobacterium damselae* subsp. *piscicida* | SaL11_281117.6 | round sardinella | Liver | Kishon estuary |
| OP493096 | *Photobacterium damselae* subsp. *Damselae* | SaL2_191017.1 | round sardinella | Liver | Kishon estuary |
| OP493097 | *Photobacterium damselae* subsp. *Damselae* | SaL3_191017.21 | round sardinella | Liver | Kishon estuary |
| OP493098 | *Photobacterium damselae* subsp. *piscicida* | SaL4_191017.12 | round sardinella | Liver | Kishon estuary |
| OP493099 | *Photobacterium damselae* subsp. *piscicida* | SaL6_191017.17 | round sardinella | Liver | Kishon estuary |
| OP493100 | *Photobacterium damselae* subsp. *piscicida* | SaL7_191017.4 | round sardinella | Liver | Kishon estuary |
| OP493101 | *Photobacterium damselae* subsp. *piscicida* | SaL8_191017.6 | round sardinella | Liver | Kishon estuary |
| OP493102 | *Photobacterium damselae* subsp. *piscicida* | SaL9_211117.22 | round sardinella | Liver | Acre |
| OP493179 | *Photobacterium damselae* subsp. *piscicida* | SuK2_140917B9 | brushtooth lizardfish | Kidney | Kishon estuary |
| OP493178 | *Photobacterium damselae* | SuL1_140917A3 | brushtooth lizardfish | Liver | Kishon estuary |
| OP493149 | *Photobacterium damselae* | UmK1_281117.15 | goldband goatfish | Kidney | Kishon estuary |
| OP493147 | *Photobacterium damselae* | UmL1_281117.22 | goldband goatfish | Liver | Kishon estuary |
| OP493148 | *Photobacterium damselae* | UmL2_281117.20 | goldband goatfish | Liver | Kishon estuary |

## 4. Discussion

The overarching aim of this research was to study the prevalence of *P. damselae* along the Israeli shoreline and determine if it correlates with different levels of pollution. We have found prevalence of *P. damselae* in all four wild fish species tested, though in varying percentages. Of 334 samples, we observed 47 positive samples for *P. damselae*, an overall prevalence of 14%, and a finding on par with estimates based on studies performed in the Mediterranean and worldwide [46–48]. Of these 47 *P. damselae*-positive samples, 29/47 (62%) were collected in the summer and 18/47 (38%) during winter. That coincides to some degree with other studies' data [49,50]. The highest prevalence was found in the round sardinella, 16/80 (20%). This may be because, in contrast with the other fish species sampled which live at depths of 30–300 m, this species is thermophilic and dwells also near beaches [51], and perhaps also, the difference may be attributed to this species' high dietary plasticity [52], or its preference for clear saline water with a minimum temperature of 24 °C [53], conditions that represent the ideal physiological temperature for the spread of *P. damselae*. It may also result from phenotypic differences between fish species in terms of susceptibility to pathogens. We also hypothesize that higher prevalence in this species may be related to the level of pollution in its environment, as it is well known that a polluted environment weakens the immune system of fish, while at the same time a high bacterial load increases chances of infection [54–56]. If the case is that a source of pollution is land-originated, then it would be expected that fish living near the coast would be more affected by it, as appears to be so with the data emerging from this research.

If so, round sardinella may be regarded as a bioindicator (sentinel) for near-shore pollution. If verified, incidents of *P. damselae* in round sardinella may provide a tool for future studies, enabling the creation of spatial mapping of pollutants, aiding in the pinpointing of their origins, whether land-originated or further and deeper at sea, bearing significance to the fish populations and public health. Because of the importance of identifying subspecies of *P. damselae*, we used PCR primers from earlier studies for preliminary identification [29,57]. Then, we used other primers to identify the subspecies of the samples found to be *P. damselae*-positive. Apparently, and as emerges from other studies [28], an efficient PCR-based assay of differentiating between the two subspecies has yet to be developed (we managed to identify only 47% of them). Perhaps the solution lies in a combination

of more advanced molecular methods such as amplified fragment length polymorphism (AFLP) together with PCR, as detailed in several studies, or other classification methods relying on phenotypic traits and bacterial culture [12,28]. Nonetheless, the fact that 20 of the 22 *P. damselae* samples that could be identified to subspecies were found to be *P. damselae* subsp. *piscicida* is noteworthy.

The results of this study present a similarity in pathogen prevalence in livers and kidneys (15.4% and 12.6%, respectively), a finding somewhat contradictory with those of Meron and colleagues (2020), in which significantly higher prevalence in the kidneys was detected [11]. Furthermore, our results were attributed by a single-organ infection, meaning all 47 infected samples came from 47 individual fish. Fish were infected in either kidney or liver, but not in both. We could not infer the meaning of this finding, but this discrepancy warrants further comparative research and speaks for the practice of collecting samples from both organs.

The results of this study showed no prevalence of RLOs in any of the 334 samples tested. Perhaps due to the importance of this pathogen in aquaculture, to date, most of the reported outbreaks of RLOs were in salmon-rearing farms, and much less from wild populations [12,14]. Reports of this pathogen from non-salmonid fish species emerged during the mid-1990s, coming first from the Pacific US, and later from farms around the Mediterranean Sea, all cases in white seabass, *Atractoscion nobilis* [33]. It should be noted that the cause of disease was usually discovered after main clinical signs including unnatural swimming patterns, loss of appetite, and lethargy [33,58,59]. In light of these data, it is not surprising that this research, being focused on asymptomatic wild fish, which have never been reported to be infected with RLOs, has resulted in no evidence of RLO infection. To increase chances of finding this pathogen in Mediterranean fishes, a survey of cultured fish should be carried out in individuals reported to express similar clinical signs or unexplained mortality. Aside from external signs on the skin, it is known that internal organs such as the liver, kidneys and spleen are affected [60]. Nevertheless, future studies may benefit from adding histological transects, looking at tissue under light- and electron microscopes and sampling the brain as well, as isolation of RLOs from brain samples was reported [33]. Such a practice may lead to increased detection sensitivity and chances of retrieving positive results.

In this research we have focused on four common food fishes that were sampled at five sites along the coast of Israel. Although all species studied are not at eminent risk of extinction (i.e., Least Concern), the four species have varying commercial value in the Mediterranean Sea, pointing to the need of closer monitoring not only for fish stocks considering exploitation levels, but also the health status of populations—as diseases caused by pathogens can lead to population declines, particularly in a climate-change scenario. This is because the potential for dispersing pathogens can increase significantly with an increase in water temperature [61], as well as increased bioavailability of pollutants in coastal areas [62]. A point of concern is, disease outbreaks and enhanced pathogen spread potential might lead to both economic and conservation issues, as diseased fish cannot be marketed, and might also have their reproductive potential reduced, which can alter populations' trajectory and cause significant loss of parental biomass. That said, constant monitoring of pathogens percentages of prevalence is needed, not only from a sanitary/economic, but also from a conservationist perspective.

We found that the prevalence of *P. damselae* was 2.6 times higher in the northern sampling sites compared with the central-southern ones. The difference in prevalence may be related to chief pollution sources along the coastline, such as those granted permission to release effluent to the sea, e.g., industrial sites, desalination and power plants, and wastewater treatment facilities. Other pollution sources include estuaries, in which higher levels of pollution have been detected in comparison to those found along the coastal plains. This is probably the result of wastewater being dumped into those rivers, suffering from low natural flow other than winter flooding [63]. According to data presented by the Ministry of Environmental Protection, Israel's northern estuaries are more polluted than the southern

ones [63]. Similarly, in the case of ports and marinas, which are sources of coastal pollution, the ports of Haifa-Carmel show higher levels of pollution than the port of Ashdod [63]. Surveys conducted on terrestrial animals found similar trends in the area of Haifa's port and the northern shores compared to those in the south [63]. A survey conducted in 2016 as part of the Israeli National Monitoring Program in the Mediterranean [63], studied the presence of metals in fish and included the striped red mullet and the goldband goatfish. In similar fashion to the findings of soil pollution and in line with the results obtained in this research, the abnormalities were found in fish collected from Acre and Haifa (northern coast). However, according to the report, "Pollutant loads in the sea—2016" [64], published by the National Unit for Marine Environment Protection (of the Ministry of Environmental Protection), the biggest source of pollution in Israel is the sludge originating from the Shafdan (the wastewater treatment plant of central Israel). This likely elevated pollution in the central-southern shores. Discharge of the sludge is carried out at a depth of 38 m, some 5 km out at sea; therefore, it could have been expected that elevated pathogen prevalence would have been observed in various fish species typically found further away from the coasts relative to those residing closer to shore. According to "Pollutant loads in the sea—2017" [65], in February 2017 several sources of wastewater and effluent (Acre, Nahariya and Shafdan) have ceased discharge into the sea, while at the same time, several new sources of pollution were added, such as natural gas drilling (northern shores), and the number of discharge permits was increased. The discharge is performed directly, indirectly via the Kishon stream or through brine terminuses (Acre, Shafdan). The fish in this study were collected between August 2016 and December 2017. The results strengthen the interpretive that fish dwelling in a polluted area are capable, under certain conditions, to take up pollutants such as metals, organic matter and pathogenic microorganisms [54,55]. Future studies should consider sorting the results based on additional factors such as temperature and salinity. From this, high importance to public health can be deduced. Based on these findings, we recommend broadening the research to create a quantitative index that will inform taking measures of precaution—the banning of fishing in significantly or potentially polluted areas. To the best of our knowledge, there is no lawful authority in Israel to impose a ban of fishing in polluted areas.

## 5. Conclusions

This study was the first to describe the prevalence of two important fish pathogens in marine fish from the eastern Mediterranean Sea. We demonstrated that *Rickettsia*-like organisms probably do not pose a risk to fish health in our region, while *Photobacterium* species were indeed present in the tested marine fishes. Further investigations are needed for understanding the impact of this zoonotic potential pathogen on local mariculture and the level of threat to public health.

**Author Contributions:** Conceptualization: D.M., Y.M., D.T. and S.H.; Data curation: Y.M., N.D. and Y.N.-B.; Formal analysis: Y.M., Z.Z.-S., Y.N.-B., P.I. and N.W.; Funding acquisition: D.M. and D.T.; Investigation: Y.M., N.D., Z.Z.-S., N.W. and S.H.; Methodology: Y.M., N.D., Z.Z.-S., Y.N.-B., P.I., N.W. and S.H.; Project administration: D.M., Y.N.-B. and S.H.; Resources: D.M., D.T. and S.H.; Supervision: D.M., D.T. and S.H.; Validation: D.M., N.D. and Y.N.-B.; Visualization: P.I.; Writing—original draft: D.M., Y.M., N.D., Z.Z.-S., Y.N.-B., P.I., N.W., D.T. and S.H.; Writing—review & editing: D.M., N.D., Z.Z.-S., P.I. and N.W. All authors have read and agreed to the published version of the manuscript.

**Funding:** This study was financially supported by the Kahn Foundation and by the Hong Kong Branch of Southern Marine Science and Engineering Guangdong Laboratory (Guangzhou, China) (Grant number SMSEGL20SC02).

**Institutional Review Board Statement:** Ethical review and approval were waived for this study, due to the fact that the fish sampled were acquired dead from fishermen/merchants in markets near the fishing grounds, as stated in the Fish collection section of the Methods.

**Informed Consent Statement:** Not applicable.

**Data Availability Statement:** Data is contained within this article. Any additional data will be provided upon request from the corresponding author.

**Conflicts of Interest:** The authors claim no competing financial interests or any other conflict of interest.

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
