# Peer review of "Molecular Identification of Photobacterium damselae in Wild Marine Fish from the Eastern Mediterranean Sea"

_fishes, doi:10.3390/fishes8020060_

Round 1
Reviewer 1 Report
The manuscript "Molecular Identification of Photobacterium damselae and Rickettsia-like Organisms (RLOs) in Wild Marine Fish from the Mediterranean Sea" (manuscript ID: fishes-2031371) describes the analyzes carried out on several organs of four different Mediterranean fish species to highlight the presence of two groups of bacterial pathogens
The article is clear and well written. For this reason, the article is recommended for publication following some minor corrections.
- Check the correct formatting of the references in the manuscript (e.g. correct in lines 134, 165 and 171).
- Line 103: write Rickettsia in italics.
- Chapter 2.4: Were the PCR conditions used the same for each protocol or are those not indicated available in the cited articles? Briefly, specify this aspect.
- Table 2: review the table indicating all the primers used in the study with their relative information. Indeed, it is not easy for the reader to summarize the information included between the table and chapter 2.4.
- Line 191: write P. damselae in italics.
- Line 200: indicate the complete name of OR.
- Line 241: delete the sentence "RLOs are a group of species, some of which are not described".
Author Response
The manuscript "Molecular Identification of Photobacterium damselae and Rickettsia-like Organisms (RLOs) in Wild Marine Fish from the Mediterranean Sea" (manuscript ID: fishes-2031371) describes the analyzes carried out on several organs of four different Mediterranean fish species to highlight the presence of two groups of bacterial pathogens
The article is clear and well written. For this reason, the article is recommended for publication following some minor corrections:
- Check the correct formatting of the references in the manuscript (e.g. correct in lines 134, 165 and 171).
Authors' Response (AR): corrected (although proper formatting is unclear for this kind of referencing).
- Line 103: write Rickettsia in italics.
AR: done. We went over the entire manuscript to look for more occurrences.
- Chapter 2.4: Were the PCR conditions used the same for each protocol or are those not indicated available in the cited articles? Briefly, specify this aspect.
AR: PCR conditions now specified in the text (now rows 171-175).
- Table 2: review the table indicating all the primers used in the study with their relative information. Indeed, it is not easy for the reader to summarize the information included between the table and chapter 2.4.
AR: This is a good point, thanks. We have reviewed this whole section (2.4) and Table 2, and hope that it is all much clearer now.
- Line 191: write P. damselae in italics.
AR: done. We went over the entire manuscript to look for more occurrences.
- Line 200: indicate the complete name of OR.
AR: done (now appears in row 206).
- Line 241: delete the sentence "RLOs are a group of species, some of which are not described".
AR: done.
Reviewer 2 Report
In this study, 334 fish samples from five fishing sites in Israel were screened for Photobacterium damselae and Rickettsia-like organisms (RLOs). 47 were found to be P. damselae-positive and none of the samples were found positive for RLOs. In general, this manuscript is not clear what the significance of the study proposed truly is. As a epidemiological study, it is suggest that author could increase the phylogenetic tree analysis among different isolates and detect more virulence genes of P. damselae strains. If this study focuses on the relationship between pathogens and the environment, it is suggest to increase the experimental data about environmental indicators of the sampling point. It need for additional work that would strengthen the authors' conclusions and arguments.
Author Response
In this study, 334 fish samples from five fishing sites in Israel were screened for Photobacterium damselae and Rickettsia-like organisms (RLOs). 47 were found to be P. damselae-positive and none of the samples were found positive for RLOs. In general, this manuscript is not clear what the significance of the study proposed truly is. As a epidemiological study, it is suggested that author could increase the phylogenetic tree analysis among different isolates and detect more virulence genes of P. damselae strains. If this study focuses on the relationship between pathogens and the environment, it is suggested to increase the experimental data about environmental indicators of the sampling point. It need for additional work that would strengthen the authors' conclusions and arguments.
AR: Thank you for these comments. We agree that additional experimental data would increase the strength of our results but unfortunately, we were unable at this specific samples' collection to retrieve more biological samples, due to the fact that the study was done in collaboration with different research groups, each requiring similar samples as well. Regarding the phylogenetic analysis - we did the best we could with our attempts to speciate Photobacterium into its species and subspecies, but alas, we could not deepen it further and delve into specific virulence genes. We believe that this study, similar to other studies we recently published (see references below) [1–4] are indeed important for the beginning of our understanding regarding pathogens in the marine environment in our region.
Reviewer 3 Report
Comments fishes-2031371
Manuscripts fishes-2031371 provides the current prevalence of pathogenic bacteria (Photobacterium damselae and Risckettsia-like organisms, RLOs) in wild fish collected from Mediterranean Sea based on molecular identification. However, the current form of ms is not suitable for publication due to the existing issues in the text. The quality of ms needs to be improved by addressing the following comments:
Abstract
Line 20 - divided into
22 - ……was done using PCR analysis with specifically designed primers on DNA extracted from livers and kidneys
29-32 - modify and simplify the sentence!
Introduction
38 - Fishes are
42-6 - modify the sentence!
61-3 - P. damselae
88-91 - modify the sentence!
91 - introduce ureC gene in the text! Why did the authors chose to use this gene to identify bacterial species in this study?
Materials and methods
- In addition to molecular approach, It would also be better to include the morphological and physiological characterizations to identify bacterial pathogens in the present study!
139-41 - include the Latin name of fish species mentioned here!, then use their common names in Table 1 instead of the Latin name. Apply this to all fish species mentioned in the text!
169-72 - please refer to the previous comment on line 91!
173-4. This sentence could be combined to the previous paragraph!
182 - state the dye used for DNA staining on the gel!
187 - Table 2: a) Double check the sequence primers of ureC and toxR, they are completely same, maybe typing errors, b) should be 16S rRNA!, c) what is toxR? explain it in the text!, and d) a modified primer should be also included in the Table 2!
Results
- Figures 1 & 2, and Table 3 - see the previous comment regarding the fish species names
204 - P. damselae
228 - was higher
Discussion
240-1 - rewrite these sentences
242 - P. damselae
251-2 - Do the authors have examples showing the correlation of fish phenotypic with susceptivity to pathogens? State it
263 - was
318 - were
323 - showed
335-6 - include references for this statement!
341-6 - this part is only repetition from Introduction section
346-51- these statements are not necessary to be included here, and can be omitted!
- Conclusion section should be included!

Author Response
Manuscripts fishes-2031371 provides the current prevalence of pathogenic bacteria (Photobacterium damselae and Risckettsia-like organisms, RLOs) in wild fish collected from Mediterranean Sea based on molecular identification. However, the current form of ms is not suitable for publication due to the existing issues in the text. The quality of ms needs to be improved by addressing the following comments:
Abstract
Line 20 - divided into
AR: changed accordingly.
22 - ……was done using PCR analysis with specifically designed primers on DNA extracted from livers and kidneys
AR: changed accordingly.
29-32 - modify and simplify the sentence!
AR: done. Good comment, thanks.
Introduction
38 - Fishes are
AR: changed accordingly.
42-6 - modify the sentence!
AR: sentence modified.
61-3 - P. damselae
AR: changed accordingly
88-91 - modify the sentence!
AR: done
91 - introduce ureC gene in the text! Why did the authors chose to use this gene to identify bacterial species in this study?
AR: inserted (Introduction, rows 97-98)
Materials and methods
- In addition to molecular approach, It would also be better to include the morphological and physiological characterizations to identify bacterial pathogens in the present study!
AR: Thank you for this comment and we absolutely agree that further characterization would improve this manuscript, but due to different technical limitations, this study was designed as a molecular study only, and did not include any classic bacteriology in the form of bacteria cultivation. Being so, it's something we can't do retroactively.
139-41 - include the Latin name of fish species mentioned here!, then use their common names in Table 1 instead of the Latin name. Apply this to all fish species mentioned in the text!
AR: changed accordingly.
169-72 - please refer to the previous comment on line 91!
AR: okay.
173-4. This sentence could be combined to the previous paragraph!
AR: combined.
182 - state the dye used for DNA staining on the gel!
AR: We used EtBr (row 184).
187 - Table 2: a) Double check the sequence primers of ureC and toxR, they are completely same, maybe typing errors, b) should be 16S rRNA!, c) what is toxR? explain it in the text!, and d) a modified primer should be also included in the Table 2!
AR: thanks for pointing this out, indeed there was a typing mistake and now the ureC sequence is correct. Furthermore, we have added some explanation regarding toxR (rows 99-103).
Results
Figures 1 & 2, and Table 3 - see the previous comment regarding the fish species names
AR: changed accordingly
204 - P. damselae
AR: changed accordingly, throughout the manuscript.
228 - was higher
AR: changed accordingly.
Discussion
240-1 - rewrite these sentences
AR: done.
242 - P. damselae
AR: changed
251-2 - Do the authors have examples showing the correlation of fish phenotypic with susceptivity to pathogens? State it
AR: A correlation between fish phenotypes and pathogen susceptibility is a great question, beyond the scope of this research. Nonetheless, intrigued by this point, we went out to search for studies on the topic and learned how complex and multifaceted this subject is. For example, one may look at the genetics of closely related lines of fish, one showing higher susceptibility than the other – and search for the genetic differences [5,6]; while another study may be looking at fish microbiota of fish sharing the same environment (usually in the guts, but the skin and gills are also good organs to compare) [7]; and yet other researchers may be examining the pathogens themselves and searching for differences in virulence genes [8].
Having all that been said, we demonstrated in Figure 1 the correlation between fish species and susceptibility to Photobacterium damselae, but could provide only hypotheses regarding these differences.
263 - was
AR: sentence slightly modified (row 273).
318 - were
AR: changed accordingly
323 - showed
AR: changed accordingly
335-6 - include references for this statement!
AR: reference added.
341-6 - this part is only repetition from Introduction section
AR: deleted
346-51- these statements are not necessary to be included here, and can be omitted!
AR: deleted
- Conclusion section should be included!
AR: Conclusion added
Reviewer 4 Report
REVIEW OF THE ARTICLE BY DANNY MORICK ET AL. ENTITLED “MOLECULAR IDENTIFICATION OF PHOTOBACTERIUM DAMSELAE AND RICKETTSIA-LIKE ORGANISMS (RLOS) IN WILD MARINE FISH FROM THE MEDITERRANEAN SEA” (fishes-2031371)
The authors detected the strains of the pathogenic bacterium Photobacterium damselae (Gammaproteobacteria, Vibrionaceae) in the Mediterranean fishes based on primary 16S rRNA gene screening by PCR and then by sequencing. They worked with large fish datasets. A bit confusing part of the work is an attempt to detect Rickettsia-like pathogens (RLOS) and conclude that they were absent in all the samples. These results are not representative. P. damselae is important as a fish pathogen. In general, Photobacterium spp. are important for fish and its study is a valuable matter. The article is in scope of the journal, but should be substantially revised. There are five main criticisms (see below) as well as minor points.
-
References. References are too old. Please, add and discuss last 3-5 year works on the abundance of photobacterium in fish, e.g. doi.org/10.1186/s40168-019-0681-y, doi.org/10.1016/j.aquaculture.2020.736175, 10.1016/j.jphotobiol.2020.111895, especially in Mediterranean Sea, doi.org/10.1007/s11356-022-22752-z.
-
Results description. Results description is verbal. Particularly, the data of electrophoresis played a key role in the work that should be given and described. Representative electropherograms should be added, at least for 16S rRNA gene sequences deposited to GenBank. Fragment lengths should be indicated. Type of mass markers should be presented. Level of statistical difference should be reflected in figures. Positive and negative control should be presented.
-
SImple BLAST comparison is insufficient for subgenus identification of Photobacterium spp. Phylogenetic analysis should be presented. It should involve type strains of Photobacterium, an outgroup. Topology should be supported by the bootstrap support. The references for phylogenetic tree reconstruction, multiple alignment and recent phylogenetic revisions should be included (doi.org/10.3390/microorganisms6010024; doi.org/10.1099/ijsem.0.002325; doi.org/10.1016/j.ympev.2010.08.012).
-
It is not a good idea to discuss RLOS and make it the main part of the work. They have not been detected in the study. There are many fish pathogens. Most likely, they were also absent. I suggest removing this part of the work and focusing on Photobacterium.
-
Discussion. Is it possible to connect the results with fish diet, time of the Year and host phylogeny.
SPECIFIC COMMENTS
l. 39. What is the difference between ‘parasites’ and fungi and bacteria? Fungi and bacteria also can be parasitic.
l. 48-50. English names of fishes should not be with capital letters.
Table 2. 16S should be 16S rRNA or ssu.
l. 134-135. “Yanong (2003) (ref. 33)” should be “Yanong [33]”.
l. 144. Which parasites? It should be documented in the text.
l. 157-172. Check reference format. (Rajan et al., 165 2003). (Osorio et al., 2000).
l. 157-172. All primers should be listed in the table.
l. 184. Please, describe the procedure of sequencing (sequencer type, reagents, in both ends or not).
l. 186. What does it mean (sequences were determined )?
l. 182-186. Provide references for g Chromas®, MEGA7® and BLASTn algorithm. Which algorithm was used for sequence alignment in MEGA (l. 216). Please, provide it with a reference. How were PCR products purified from the gel?
Table 3 legend. accession numbers for which genes?
l. 240. “of these two ” - at first mention in Discussion indicate them.
l. 341-344. Repetition of Introduction. Remove. In general, results are repeated in the Discussion (first sentences of paragraphs).
Author Response
We are uploading a file consisting all of our replies to the four reviewers (attached)

Round 2
Reviewer 2 Report
The author did not add research according to the previous review. I think the work in this study is not up to the level published in Fishes.
Author Response
thanks

Reviewer 3 Report
Comments fishes-2031371_v2
Manuscript fishes-2031371 has been significantly improved by the authors according to the previous comments. However, some minor issues are still found in the revised ms. The ms could be considered for publication after addressing those remaining issues.
145-8 - the Latin names of species should come after the common names!
367 - was
369 - did not
370 -were

Author Response
Thanks for pointing out these additional minor issues.
We have fixed them in the (next) submitted ms, which will be uploaded once we finish addressing all other comments reviewers gave us.
Good day,
PI
Reviewer 4 Report
I have read the text and authors' responses. Unfortunatly main concerns were not addressed.
One of key results is identification of P. damselae subspecies. In the responses they say that "CAR-1 and CAR-2 allowed for the specific detection of P. damselae, while the lack of ureC gene suggested the isolates were in fact the subspecies piscicida (or subsp. damselae, when ureC was present).", however this fact is not reflected in the results (Table 3).
The authors says that thay, "agree it should not be a major part of the work and the word count shows that it accounts for merely ~250 words (each) in the Introduction and Discussion", however detection of RLOs is stated in the title and abstract. Thus, it is stated as a main part of the study. "The overarching aim of this research was to study the prevalence of RLOs and P. damselae" (l. 248-249) - this aim is not supported by any figures/tables as well as data on RLO detection. Verbal presentation of results not supported by figures or tables looks unscientific.
Minor points: P. damselae is not always italicized.
Author Response
Molecular Identification of Photobacterium damselae and Rickettsia-like Organisms (RLOs) in Wild Marine Fish from the Mediterranean Sea
Dear Fishes editorial board,
We would like to thank the reviewers for taking the time to read the revised manuscript and suggest some more changes in order to improve the work and make it merit of being published in this journal.
We did our best to answer all issues the reviewers mentioned in the 2nd round of the reviewal, but admit that not all suggestions could be incorporated into the manuscript, as mentioned in our previous reply.
Nonetheless, with the valuable aid of the reviewers, it is now better presented to the readers.
Following are the comments made by the Reviewers and the replies we gave them:
Reviewer 2:
Comments and Suggestions for Authors
The author did not add research according to the previous review. I think the work in this study is not up to the level published in Fishes.
Authors' Reply (AR): Further experiments at this point cannot be performed. We hoped our explanations of this point would be sufficient, but understand the reviewer disagrees.
Reviewer 3:
Comments and Suggestions for Authors
Comments fishes-2031371_v2
Manuscript fishes-2031371 has been significantly improved by the authors according to the previous comments. However, some minor issues are still found in the revised ms. The ms could be considered for publication after addressing those remaining issues.
145-8 - the Latin names of species should come after the common names!
AR: fixed.
367 - was
AR: fixed.
369 - did not
AR: fixed.
370 -were
AR: fixed.
Reviewer 4:
I have read the text and authors' responses. Unfortunatly main concerns were not addressed.
One of key results is identification of P. damselae subspecies. In the responses they say that "CAR-1 and CAR-2 allowed for the specific detection of P. damselae, while the lack of ureC gene suggested the isolates were in fact the subspecies piscicida (or subsp. damselae, when ureC was present).", however this fact is not reflected in the results (Table 3).
AR: from what we understand, the reviewer was concerned that Table 3 would not be self-explanatory of the way the sequences were identified, and/or the fact that 5 of the 18 sequences were not sub-speciated, but rather just listed as P. damselae. We therefore made an addition after the table, reminding the reader of the process portrayed in section 2.4 (PCR), and stressing out the results. We hope this is sufficient.
The authors says that thay, "agree it should not be a major part of the work and the word count shows that it accounts for merely ~250 words (each) in the Introduction and Discussion", however detection of RLOs is stated in the title and abstract. Thus, it is stated as a main part of the study. "The overarching aim of this research was to study the prevalence of RLOs and P. damselae" (l. 248-249) - this aim is not supported by any figures/tables as well as data on RLO detection. Verbal presentation of results not supported by figures or tables looks unscientific.
AR: we have removed the term RLOs from the title, abstract, beginning of the discussion and from the key words, and added a sentence in the beginning of the part explaining about RLOs within the introduction, stressing that this was a secondary objective of the study. we hope that now RLOs do not seem like a major part of the study, but can still be mentioned thus in the ms.
Minor points: P. damselae is not always italicized.
AR: fixed.

Round 3
Reviewer 4 Report
The article can be accepted if the Editor(s) feel it reaches quality of the journal.
Author Response
We are deeply gratified for this reviewer's efforts to go over the ms and provide all the valuable suggestions and remarks aimed at improving it.
The reviewer still thinks the presentation of the Results and the support they provide the conclusion can be improved.
We would be happy to follow any suggestions on this, which would also help us understand why this reviewer is still not satisfied with how the ms can be further improved.
However, since none were given, we are hoping the editors still find the ms in its current form as suitable for publication.